# Protective Immunity of COVID-19 Vaccination with ChAdOx1 nCoV-19 Following Previous SARS-CoV-2 Infection: A Humoral and Cellular Investigation

**DOI:** 10.3390/v14091916

**Published:** 2022-08-30

**Authors:** Tamiris Azamor, Ingrid Siciliano Horbach, Danielle Brito e Cunha, Juliana Gil Melgaço, Andréa Marques Vieira da Silva, Luciana Neves Tubarão, Adriana de Souza Azevedo, Renata Tourinho Santos, Nathalia dos Santos Alves, Thiago Lazari Machado, Jane Silva, Alessandro Fonseca de Souza, Camilla Bayma, Vanessa Pimenta Rocha, Ana Beatriz Teixeira Frederico, Brenda de Moura Dias, Bruno Pimenta Setatino, Caio Bidueira Denani, Samir Pereira da Costa Campos, Waleska Dias Schwarcz, Michel Vergne Sucupira, Edinea Pastro Mendes, Edimilson Domingos da Silva, Sheila Maria Barbosa de Lima, Ana Paula Dinis Ano Bom, Sotiris Missailidis

**Affiliations:** Bio-Manguinhos/Oswaldo Cruz Foundation, Rio de Janeiro 21040-900, Brazil

**Keywords:** SARS-CoV-2 vaccine, ChAdOx1 nCoV-19, hybrid immunity, cellular and humoral immunity

## Abstract

Infections caused by SARS-CoV-2 induce a severe acute respiratory syndrome called COVID-19 and have led to more than six million deaths worldwide. Vaccination is the most effective preventative measure, and cellular and humoral immunity is crucial to developing individual protection. Here, we aim to investigate hybrid immunity against SARS-CoV-2 triggered by the ChAadOx1 nCoV-19 vaccine in a Brazilian cohort. We investigated the immune response from ChAadOx1 nCoV-19 vaccination in naïve (noCOVID-19) and previously infected individuals (COVID-19) by analyzing levels of D-dimers, total IgG, neutralizing antibodies (Nabs), IFN-γ (interferon-γ) secretion, and immunophenotyping of memory lymphocytes. No significant differences in D-dimer levels were observed 7 or 15 days after vaccination (DAV). All vaccinated individuals presented higher levels of total IgG or Nabs with a positive correlation (R = 0.88). Individuals in the COVID-19 group showed higher levels of antibody and memory B cells, with a faster antibody response starting at 7 DAV compared to noCOVID-19 at 15 DAV. Further, ChAadOx1 nCoV-19 vaccination led to enhanced IFN-γ production (15 DAV) and an increase in activated T CD4+ naïve cells in noCOVID-19 individuals in contrast with COVID-19 individuals. Hence, our data support that hybrid immunity triggered by ChAadOx1 nCoV-19 vaccination is associated with enhanced humoral response, together with a balanced cellular response.

## 1. Introduction

Since the World Health Organization (WHO) declared COVID-19 a public health emergency at the beginning of 2020, the SARS-CoV-2 virus has been responsible for approximately 6,250,000 deaths worldwide as of May 2022 [1,2].. Cellular and humoral immunity is crucial for positive disease outcomes and the development of protective immunity after recovery from COVID-19. It was observed that early, robust T cell and type I interferon responses are needed for viral clearance and to attenuate disease severity [3,4]. Further, memory CD4+ and CD8+ T cells secreting interferon-γ (IFN-γ) occur at increased levels during the convalescent phase and are associated with complete recovery [5]. Despite this, T cells could be correlated with poor clinical outcomes due to high activation, exhaustion, proliferation, and peripheral reduction of cytotoxic CD8+ T cells [6,7,8]. Concerning humoral immunity, several studies described the total decay of IgG anti-SARS-CoV-2 spike protein levels in the months following COVID-19, allowing for future reinfection [9,10,11,12]. This decrease in total antibody levels is partially offset by a per-antibody increase in neutralizing titer and subtype, as well as induction of long-lasting memory T and B cells. This immunological landscape avoids hospitalization and deaths, and the same outcome is expected after the use of COVID-19 vaccinations [12,13,14].

The knowledge acquired with previous emergences of SARS-CoV-1 and MERS-CoV helped in the rapid development of SARS-CoV-2 vaccines. A diverse variety of vaccines have been developed, including a DNA-based vaccine by Yu et al. based on the live attenuated YF17D-vector from Sanchez-Felipe, as well as the mRNA-based vaccines by Pfizer-BioNTech (BNT162b2) and Moderna (mRNA-1273) [15,16,17,18]. One of the vaccines created was ChAadOx1 nCoV-19 (AZD1222), developed at the University of Oxford by combining a codon-optimized full-length Wuhan SARS-CoV-2 spike protein gene (GenBank accession number YP_009724390.1 ) with the ChAdOx1 vector [19]. This vector is a replication-deficient chimpanzee adenovirus without reactogenicity that was previously utilized to protect non-human primates (NHPs) against MERS-CoV-induced disease [20,21,22]. Many clinical trials using homologous prime–boost doses 8–12 weeks apart analyzed samples from SARS-CoV-2 naïve individuals to investigate their cellular and humoral responses after vaccination. These investigations included quantitation of IFN-γ-secreted cells, total anti-spike, and neutralizing antibodies, which are considered the standard analytes for SARS-CoV-2 vaccine efficacy [23]. Studies using populations from the UK and India demonstrated that a single dose of the AstraZeneca COVID-19 vaccine induced polyfunctional antibodies capable of mediating virus neutralization while driving other antibody-dependent effector functions. This includes antibody-dependent neutrophil/monocyte phagocytosis and an innate and potent T cell response. However, after the second dose, despite an increase in the magnitude of humoral response, complement activation, and natural killer cell activation, the T cell response remained sustained in the individuals [23,24]. Further, Ewer et al. explored the cellular response profile eight weeks after a single dose of the ChAdOx1 nCoV-19 vaccine, which displayed elevated secretion of IFN-γ and tumor necrosis factor-α (TNF-α), specifically by CD4+ T cells [25].

Regarding ChAadOx1 nCoV-19 vaccine safety, clinical trials showed rare serious adverse reactions and self-limiting adverse events [26]. In addition, it was demonstrated that ChAdOx1 nCoV-19 vaccination can result in development of the rare immune condition thrombotic thrombocytopenia, which is mediated by platelet-activating antibodies against PF4 with elevated D-dimer levels [27,28].

It remains elusive how SARS-CoV-2 infection prior to vaccination could disrupt the amount of total anti-SARS-CoV-2 spike neutralizing antibodies and IFN-γ secretion levels elicited by the ChAadOx1 nCoV-19 vaccine [29,30]. With the current global COVID-19 situation presenting a sustained number of cases, mostly mild due to vaccination efforts, it is imperative to study hybrid immunity against SARS-CoV-2. For this reason, we aim to analyze vaccine immunogenicity and possible D-dimer alterations in a Brazilian cohort with or without previous SARS-CoV-2 infection prior to vaccination. These data deepen immunological knowledge regarding ChAadOx1 nCoV-19 vaccination and contribute to procedural public health development.

## 2. Materials and Methods

### 2.1. Study Design

All procedures for attaining samples were approved by the Ethics Committee of Fiocruz (CAAE: 34728920.4.0000.5262) in accordance with the Helsinki Declaration as revised in 2013 [31]. Informed consent was obtained from all participants, and all methods were carried out in accordance with relevant regulations and guidelines.

### 2.2. Participants

Our study made use of the ongoing prospective cohort study of adult workers at the Brazilian National Institute of Technology in Immunobiologicals (Bio-Manguinhos/Fiocruz) in Rio de Janeiro, Brazil. Since March 2020, 95 healthy adults have been monitored biweekly using RT-qPCR tests performed on naso-oropharyngeal swab collections for SARS-CoV-2 detection. Individuals were immunized with two doses of ChAdOx1 nCoV-19 vaccine and at least one blood sample was collected at the different time points listed below. Exclusion criteria from the study included immunization with another brand of COVID-19 vaccine, any clinical signs or symptoms 30 days prior to vaccination (e.g., fever, pains, fatigue, or cough), or a known chronic infectious disease (e.g., Hepatitis C or human immunodeficiency virus infection). The study design included blood sample collections at 0, 7, 15, 30, 90, and 120 days after vaccination (DAV), with the first dose of ChAdOx1 nCoV-19 at 0 DAV and the second dose at 90 DAV. A total of 72 participants completed the follow-up blood collections for all six time points, and 23 participants skipped at least one time point. Finally, we clustered the participants who tested positive for COVID-19 by either RT-qPCR or anti-SARS-Cov-2 serology prior to vaccination in a group titled “COVID-19” (previous COVID-19, n = 37). Those without prior history of COVID-19 were placed in a group titled “noCOVID-19” (no previous COVID-19, *n* = 58) (Table 1).

### 2.3. Samples

Between January and September 2021, blood samples were obtained from participants vaccinated with ChAdOx1 nCoV-19 in Rio de Janeiro, Brazil. Samples were collected at 0, 7, 15, 30, 90, and 120 DAV, with the first vaccine dose given at 0 DAV and the second dose given at 90 DAV. To obtain peripheral blood mononuclear cells (PBMCs) and plasma samples, blood samples were processed using a Histopaque density gradient (Sigma-Aldrich, Taufkirchen, Germany) according to the manufacturer’s suggestions, with the lysis of residual red blood cells performed using an ACK solution (Invitrogen/Thermo Scientific, Waltham, MA, USA). Isolated PBMCs were frozen in Cryostor CS5 (StemCell Technologies, Vancouver, QC, Canada) for later analysis. Individual blood samples were collected before vaccination (0 day = pre-immunization) to determine a baseline immunity profile. Blood samples used in PRNT assays were separated by centrifugation and sera were collected, inactivated (56 °C for 30 min), and stored at –30 °C. Positive and negative serum samples for Nab against SARS-CoV-2 were selected from donors with qPCR positive and negative results for SARS-CoV-2, respectively. These positive and negative serum samples were used as reference controls. The time points for sample collection and the techniques used for analysis are summarized in Table 2.

### 2.4. D-Dimer Quantitation

Plasma samples from 0, 7, and 15 DAV were used in duplicate to quantify D-dimer levels expressed in pg/mL using the Human Luminex Discovery Assay kit (R&D Systems, Minneapolis, MI, USA) according to the manufacturer’s instructions. Mean Intensity Fluorescence (MIF) values for each sample were interpolated with a standard curve using four-parameter sigmoidal regression in SoftMax Pro 5.4 (Molecular Devices, Silicon Valley, CA, USA).

### 2.5. PRNT-SARS-CoV-2 Assay

The SARS-CoV-2 virus was obtained from the inoculum of African green monkey Vero E6 cells with a clinical sample of a nasopharyngeal swab from a person positive for COVID-19 in Rio de Janeiro. This isolate was kindly provided by the Laboratory of Respiratory Viruses and Measles at IOC/Fiocruz (SISGEN A994A37). For the development of the PRNT, different amounts of plaque-forming units (PFU) were tested (50 to 120 PFU). With 60 PFU, homogeneous and individualized plaques were generated, and this amount was thus established as the standard for the assay (data not shown). The PRNT-SARS-CoV-2 assay was carried out on Vero cells grown in 24-well plates. Serum samples were serially diluted (from 1:10 to 1:31,250) followed by the addition of approximately 60 PFU of SARS-CoV-2 (2019-nCoV, Wuhan strain) and incubated for 1 h at 37 °C in a 5% CO_2_ chamber. The virus–serum mixture was added onto a confluent monolayer (200,000 cells/well, CCL-81; ATCC, USA) and incubated for 1 h at 37 °C in a 5% CO_2_ atmosphere. After incubation, the inoculum was discarded and cells were then overlaid with 1 mL of semi-solid 199 Medium supplemented with 5% fetal bovine serum (FBS) and 1.5% carboxymethylcellulose (CMC), before then being incubated for 3 days at 37 °C in 5% CO_2_ before final fixation with 1.25% formalin (*vol/vol*). Cells were stained with crystal violet, plates were photographed using the BioSpot^®^ Software Suite (CTL—Cellular Technology Limited, Shaker Heights, OH, USA), and plaques were manually counted. The PRNT50 titer was expressed as the reciprocal of the serum dilution able to neutralize the viral infection by 50%. Seropositivity rates were determined considering a serum dilution higher than 1:10 as the cut-off criterion for PRNT positivity.

### 2.6. Detection of Total Antibodies (IgG Anti-RBD)

The Promega Lumit Dx SARS-CoV-2 immunoassay (Promega, Madison, WI, USA) was used according to the manufacturer’s protocol for the detection of antibodies binding to the receptor-binding domain (RBD) of the SARS-CoV-2 spike protein. Absorbance was measured at 550 nm on a microplate reader (VersaMax-Molecular Devices, Silicon Valley, CA, USA). Results were calculated by dividing the sample relative light unit (RLU) by the mean calibrator RLU to generate a signal-to-cutoff (S/C) value. Values above 1 indicate that the sample is positive for SARS-CoV-2 antibodies.

### 2.7. IFN-γ ELISpot Assay

The frequency of IFN-γ-secreting cells in collected PBMC samples from vaccinated subjects was analyzed using a commercial human ELISpot assay (Mabtech, Cincinnati, OH, USA). PBMCs were plated (5 × 10^5^ cells/well) into pre-coated IFN-γ ELISpot plates according to the manufacturer’s protocol and cultured for 48 h in the presence or absence of 1 mg/mL of SARS-CoV-2 spike glycoprotein peptides (PepMix™) (JPT Peptides, Berlin, Germany) [29]. A positive control was set up with anti-CD3 monoclonal antibodies diluted to 1:1.000 as provided by the ELISpot commercial kit. After an initial 48 h of growth, PBMCs were washed and incubated with a biotinylated anti-IFN-γ antibody for 2 h at room temperature (RT) followed by incubation with alkaline phosphatase-conjugated streptavidin for 1 h at RT. Spots of IFN-γ-secreting cells were visualized using the 5-bromo-4-chloro-3-indolyl phosphate/nitro blue tetrazolium substrate and counted using the ImmunoSpot Image Analyzer (CTL—Cellular Technology Limited, Shaker Heights, OH, USA). Samples presenting less than 20 spots in the positive controls were excluded. After background subtraction (the number of spots formed without stimulation), the number of Spot Forming Units (SFU) per million PBMCs was calculated.

### 2.8. Activation-Induced Marker (AIM) Assay to Assess Memory Phenotypes for B and T Cells

The polyclonal antigen-specific cellular assay was performed on thawed PBMC samples grown in supplemented RPMI1640 (R10), containing 1 µg/mL of SARS-CoV-2 PepMix™ (JPT Peptides, Berlin, Germany), for 48 h in a 37 °C, 5% CO_2_ chamber. After initial incubation, cells were harvested, washed with FACS buffer, centrifuged, and stained using a viability dye (Live/Dead Fixable Blue, Thermo Fisher Scientific, Waltham, MA, USA) followed by surface staining using human antibodies: CD3 APC-Cy7, CD4 BV421, CD8 BV605, CD38 PE-Cy7, CD19 BB700, CD27 BB515, CD134 (OX40) BV650, CD197 (CCR7) BV510, and CD45RA APC. All human antibodies were purchased from BD Biosciences, San Jose, CA, USA. Two types of compensation beads were used as compensation controls (UltraCompBeads, Invitrogen, Waltham, MA, USA; ArCTM Armine Reactive Compensation Bead kit, Invitrogen, Waltham, MA, USA) [32].

The samples were acquired using LSRFortessa (BD Biosciences, San Jose, CA, USA) and analyzed with FlowJo software v.10.8 (BD Biosciences, San Jose, CA, USA). The gating strategy used for clustering memory B and T cells is represented in Appendix A. PBMCs grown in unsupplemented R10 were used as a negative control and as a measure of baseline fluorescence. The minimum percentage considered for final analysis was 1% in the gate of activated memory cells.

### 2.9. Statistical Analysis

Values of neutralizing antibodies (Nab) against SARS-CoV-2, the RLU of anti-SARS-Cov-2, and Spot Forming Units (SFU) of IFN-γ secretion were compared between groups of interest by pairwise Mann–Whitney tests. Comparisons between time points were conducted using Kruskal–Wallis tests followed by Dunn’s post-hoc tests. Correlations between antibody levels obtained from PRNT-SARS-CoV-2 and Lumit Dx SARS-CoV-2 Immunoassays were calculated using the non-parametric Spearman test. Statistical analyses were performed within GraphPad Prism software version 8.0 (GraphPad Software, San Diego, CA, USA).

## 3. Results

### 3.1. ChAdOx1 nCoV-19 Does Not Induce D-Dimer Production

It was not possible to observe significant differences when comparing D-dimer levels at any time point after ChAdOx1 nCoV-19 vaccination, regardless of previous COVID-19 status or within the complete cohort (Figure 1A,B and Appendix A). Additionally, significant differences were not observed when comparing D-dimer levels between the noCOVID-19 and COVID-19 groups (Figure 1C).

### 3.2. Analysis of the Humoral Response Elicited by ChAdOx1 nCoV-19

The humoral immune response in noCOVID-19 and COVID-19 groups was measured by the presence of total IgG against the RBD region spike protein of SARS-CoV-2 (Figure 2) or Nabs against infective SARS-CoV-2 virus (Figure 3) (Appendix A). Serum samples were collected at time points according to Table 2. All vaccinated individuals presented significantly higher levels of total IgG (Figure 2) or Nabs (Figure 3) when compared to the baseline samples (0 days = pre-immunization). However, the COVID-19 group showed a remarkable increase in the level of antibodies at 7 DAV (Figure 2B and Figure 3B) and remained elevated at all time points analyzed up to 120 DAV (Figure 2B and Figure 3B). On the other hand, the noCOVID-19 group revealed a significantly heightened amount of both total IgG and Nabs at 120 DAV compared to the other previous time points studied, indicating that a second dose of vaccine was required to significantly raise antibody levels in those individuals (Figure 2A and Figure 3A). Furthermore, the COVID-19 group elicited a significantly enhanced humoral immune response after both prime and booster doses, which was observed at all time points assessed when compared against individuals in the noCOVID-19 group (Figure 2C and Figure 3C). Total IgG anti-RBD and Nab results obtained in both groups at different time points were evaluated by Promega Lumit Dx SARS-CoV-2 immunoassays (Promega, USA) and by classical PRNT-SARS-CoV-2 and showed a positive correlation coefficient for the complete cohort (r = 0.887) (Figure 4A), the noCOVID-19 group (r = 0.809) (Figure 4B), and the COVID-19 group (r = 0.885) (Figure 4C).

### 3.3. ChAdOx1 nCoV-19 Elicits Different Profiles of Cellular Response Dependent on Pre-Vaccination COVID-19 Status

By clustering participants according to pre-vaccination SARS-CoV-2 infection, the results demonstrate that the noCOVID-19 group presented enhanced IFN-γ secretion at 15 DAV compared to 0 DAV and levels remained sustained (Figure 5A). The COVID-19 group did not present significant differences when comparing all time points after vaccination, although it was possible to observe a slight peak in IFN-γ secretion at 7 DAV with high dispersion, suggesting earlier IFN-γ secretion in the COVID-19 group compared to the noCOVID-19 group (Figure 5B).

A higher amount of IFN-γ secretion was observed at 15 DAV in the noCOVID-19 group compared to the COVID-19 group. Despite the differential IFN-γ secretion profiles observed between groups after the first dose, it was not possible to observe distinct levels of IFN-γ secretion after complete vaccination (120 DAV) (Figure 5B). Considering the complete cohort, the number of IFN-γ-secreting cells presented enhanced levels at 15 DAV and 120 DAV compared to 0 DAV, demonstrating the ChAdOx1 nCoV-19 vaccine elicited augmented IFN-γ secretion after the first and second dose (Appendix A).

Within the noCOVID-19 group, memory B cells were activated after the first and second doses of the COVID-19 vaccine (Figure 6A). A similar result was found for the activated effector memory (EM) CD4+ T cells. Activation of terminally differentiated (Temra) CD4+ T cells was seen after the second dose only (Figure 6B). In CD8+ T memory cells, the amount of activated central memory (CM) cells was significantly increased after the first dose. Meanwhile, the EM CD8+ T cells were activated after first and second doses of the vaccine. Lastly, Temra CD8+ T cells were activated after the second dose alone (Figure 6C).

In the COVID-19 group, memory B cell activation did not change with vaccination, but the percentage of activation was higher than in the noCOVID-19 group (Figure 6B). For memory CD4+ and CD8+ T cells, CM cells were activated only after the second dose, and Temra CD8+ T cells were significantly activated after the first dose (Figure 6B). It was also observed that the percentage of CM CD4+ T lymphocytes was highest in the noCOVID-19 group after the first dose when compared to individuals from the COVID-19 group (Figure 6C). The highest percentage of CM CD8+T lymphocytes was reached after the second dose for the COVID-19 group (Figure 6C), and this percentage was also higher when compared to the noCOVID-19 group. However, noCOVID-19 individuals had the highest percentages of EM CD4+ and CD8+ T cells after first and second vaccine doses. The noCOVID-19 group demonstrated the highest percentage of Temra CD8+ T cells after the second dose (Figure 6B,C).

## 4. Discussion

In the last two years, the spread of SARS-CoV-2 has been curbed by physical prevention measures such as facial masks, social distancing, and regular hand-washing. Considering the susceptibility of the host and the transmission capacity of the virus, the World Health Organization classified the different viral mutations according to the Greek alphabet, naming a Variant of Concern (VoC) for each important mutation [33,34]. Due to the rapid viral spread, SARS-CoV-2 vaccines are mostly administered to individuals already exposed to at least one SARS-CoV-2 VoC [29]. Hence, the study of hybrid protection, induced by both natural and vaccine exposure, is imperative for developing a better understanding of immunological patterns that could help create public health strategies of vaccine application, especially in underdeveloped countries. Here, our data demonstrated that, in a Brazilian population, ChAdOx1 nCoV-19 vaccine protection requires not only neutralizing and non-neutralizing antibodies, but also a balanced T cell response.

Regarding humoral response, our data exhibited that ChAdOx1 nCoV-19 vaccination elicited an enhanced response in both binding and neutralizing antibodies in previously infected individuals. Additionally, the high level of overall antibody production in the COVID-19 group can be attributed to the high percentage of memory B cell activation. Altogether, these data demonstrated the benefits of hybrid humoral immunity for SARS-CoV-2 protection and are corroborated by a previous study in populations from Malawi, India, and the United Kingdom, reinforcing the importance of regular SARS-CoV-2 testing [30,35,36]. Chibwana et al. revealed that ChAdOx1 nCoV-19 vaccination is an effective booster for waning VoC cross-variant antibody immunity after initial priming by SARS-CoV-2 infection in Malawian adults [30]. Additionally, multiple studies have reported that two doses of an mRNA-based vaccine were needed to induce peak antibody and memory B cell responses against SARS-CoV-2 in naïve individuals, with the heightened response persisting for at least for eight weeks, whereas only one vaccine dose induced peak responses in SARS-CoV-2-primed individuals [37,38,39,40]. Further, memory B cells persisted for at least three months after infection and were increased five- to ten-fold in hybrid immunity compared to vaccination alone or natural infection [39,40]. In this context, reinforcement doses were a key strategy, as demonstrated by Flaxman et al. This study showed that a third dose of ChAdOx1 nCoV-19 (28–38 weeks after the second dose) induced a strong boost in neutralizing antibody titers and activity against VoCs in individuals without prior exposure to SARS-CoV-2 [41].

Regarding T cell response, we observed that naïve individuals presented an enhanced response 15 days after initial ChAadOx1 nCoV-19 vaccination, but only maintained sustained levels after the second dose. Previous studies demonstrated the same pattern of cellular response [24,25]. From peptide stimulation of PBMCs collected 15 days after the initial ChAadOx1 nCoV-19 dose, Ewer et al. observed enhanced secretion of IFN-γ and IL-2, while IL-4 and IL-13 secretion levels were not increased [25]. Flow cytometry analysis confirmed that T CD4+ cells with a Th1 profile were increased, as expected for a vaccine response [25]. Barrett et al. demonstrated that after a second dose, despite an increase of humoral responses, complement activation, and natural killer cell activation, there was no significant increase in T cell response [24]. Further, contrary to the results observed in humoral data, here our cellular investigation demonstrated that individuals with previous SARS-CoV-2 infection did not present a higher magnitude of IFN-γ secretion or an increase in T CD8+ effector memory cells when compared to naïve individuals. However, some individuals with a previous COVID-19 history demonstrated a faster immune response at 7 DAV, suggesting an influence of *IFNG* genetic background in vaccine immunogenicity. This genetically regulated early IFN-γ response was already observed after immunization with the yellow fever (YF) vaccine 17DD, a replicative live-attenuated virus that elicits a complex Th1/Th2 response [42]. More efforts are necessary to further characterize the ChAadOx1 nCoV-19 response, although it is known that SARS-CoV-2 infections elicit a complex Th1/Th2 immunological response [43]. The need for balance in effector T cell response elicited by SARS-CoV-2 was already explored in multiple studies focusing on COVID-19 severity, demonstrating the role of lymphocyte reduction and exhaustion and its relation to final disease outcome [6].

Effector memory T cells are highly activated in the initial contact with a pathogen or vaccine combined with a high production of IFN-γ, promoting differentiation to central memory T cells which leads to long-lasting memory immunity [44]. This could explain the activation of effector memory T cells after the first dose of COVID-19 vaccine in naïve subjects compared to previously infected subjects, as observed here. Activation is sustained for both naïve and previously infected subjects after the second dose. Le Bert et al. determined that pre-existing T cells that recognized SARS-CoV proteins could be helpful in overall immunity against SARS-CoV-2, reinforcing memory T cell responses [45]. Tarke et al. showed that primary vaccination was able to induce immunological T cell memory against several variants, including Omicron, of which >84% (CD4+) and >85% (CD8+) of activated memory T cell response was detected [46]. Our findings suggest that COVID-19 vaccination promotes long-lasting memory cell production after two doses, independent of previous infection status.

In our present study, the follow-up segment loss and lack of clear cut-offs for correlates of protection against SARS-CoV-2 provided a challenge. Additionally, further investigations are necessary to characterize humoral and cellular responses against circulating VoCs elicited by the ChAadOx1 nCoV-19 vaccine, as well as the influence of previous exposure to multiple VoCs in immunological responses elicited by ChAadOx1 nCoV-19. Regardless of these limitations, here we addressed key immunological parameters, quantitation of neutralizing and binding antibodies, IFN-γ secretion by PBMCs, and immunophenotyping of memory lymphocytes demonstrating the supplemental effect of previous infection for developing protective hybrid immunity after ChAadOx1 nCoV-19 vaccination. In addition, further studies need to be carried out in order to elucidate if there are any contributions from the hybrid response to the overall duration of humoral immunity. More efforts are needed to determine protective cut-offs against SARS-CoV-2, as well as regular administration of tests for previous SARS-CoV-2 infections. Finally, our data presented here could help develop vaccine distribution schemes, especially in populations with low incomes and elevated levels of SARS-CoV-2 infections.

## Figures and Tables

**Figure 1 viruses-14-01916-f001:**
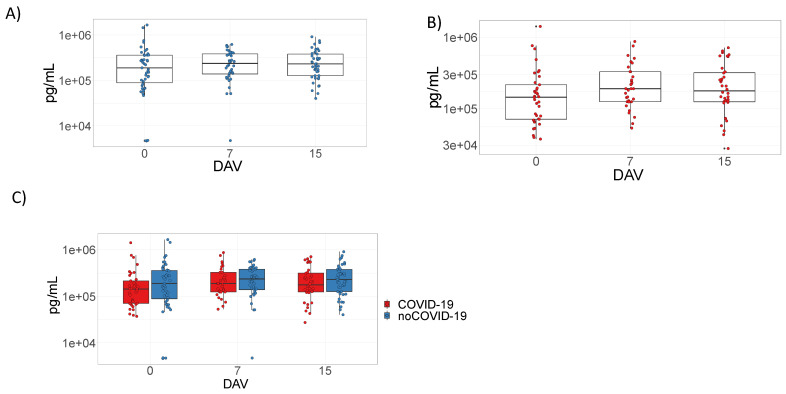
D-dimer plasma quantitation after ChAdOx1 nCoV-19 vaccination. Panels (**A**–**C**) refer to samples tested from volunteers immunized with the prime–boost protocols with ChAdOx1 nCoV-19 on days 0 and 90. Data are represented by box plots with the mean and interquartile intervals of D-dimer plasma quantitation given in pg/mL. Panels (**A**,**B**) represent comparisons between time points considering participants without pre-vaccination COVID-19 disease (noCOVID-19) (**A**) or with pre-vaccination COVID-19 (COVID-19) (**B**) using Kruskal–Wallis tests with Dunn’s multiple comparison post-hoc tests. Panel (**C**) represents comparisons between the noCOVID-19 and COVID-19 groups according to the time point analyzed using the Mann–Whitney test. The sample number varied according to follow-up segment loss. DAV = days after vaccination.

**Figure 2 viruses-14-01916-f002:**
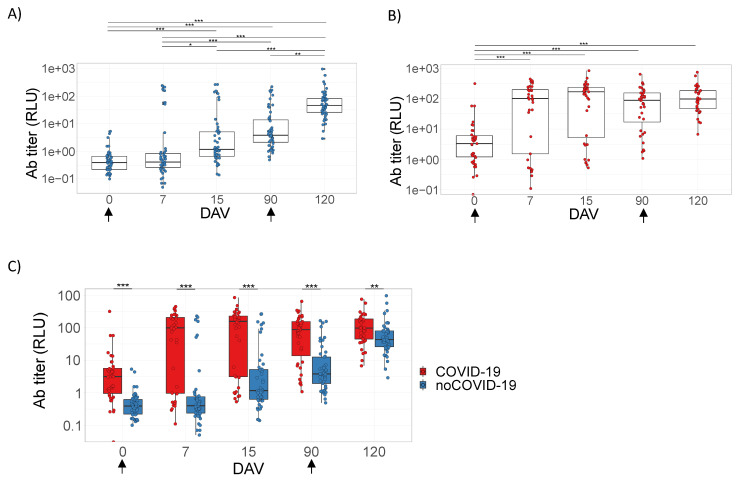
Total IgG anti-RBD (RBD region spike protein of SARS-CoV-2). Panels (**A**–**C**) refer to samples from volunteers immunized with the prime–boost protocol for ChAdOx1 nCoV-19 on days 0 and 90 (indicated by arrows). Individual serum samples were collected before vaccination (0 days = pre-immunization) to determine a baseline profile. In addition, volunteer samples were collected on days 7, 15, and 90 post-first dose and 30 days after the second immunizing dose; this point is represented on the graphs as day 120. Panels (**A**–**C**) belong to volunteer groups including individuals without (noCOVID-19) or with (COVID-19) prior SARS-CoV-2 infection and the comparison of these two groups. The ELISA was expressed by RLU (Relative Light Units). Asterisks indicate differences that are statistically significant (* *p* < 0.05, ** *p* < 0.01, and *** *p* < 0.001) as evaluated by Kruskal–Wallis tests with Dunn’s post-hoc tests (**A**, **B**) and Mann–Whitney tests (**C**). DAV = days after vaccination.

**Figure 3 viruses-14-01916-f003:**
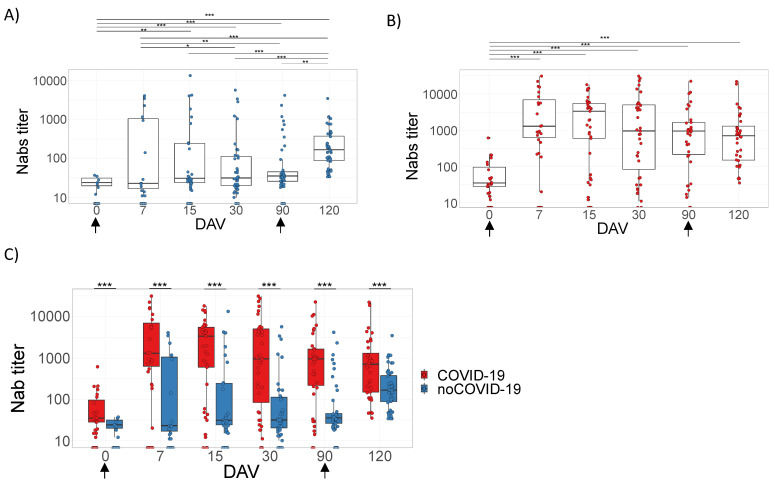
Neutralizing antibody (Nab) responses against infective SARS-CoV-2. Panels (**A**–**C**) refer to samples tested from volunteers immunized with the prime–boost protocol for ChAdOx1 nCoV-19 on days 0 and 90 (indicated by arrows). Individual serum samples were collected before vaccination (0 days = pre-immunization) to determine a baseline profile. In addition, volunteer samples were collected on days 7, 15, 30, and 90 post-first dose and 30 days after the second immunizing dose; this point is represented on the graphs as day 120. Panels (**A**–**C**) belong to volunteer groups including individuals without (noCOVID-19) or with (COVID-19) pre-vaccination SARS-CoV-2 infection and the comparison of these two groups. Sera were serially diluted from 1:10 to 1:31,250 and the PRNT50% was performed in 24-well plates. Asterisks indicate differences that are statistically significant (* *p* < 0.05, ** *p* < 0.01, and *** *p* < 0.001) as evaluated by Kruskal–Wallis tests with Dunn’s post-hoc tests (**A**,**B**) and Mann–Whitney tests (**C**). DAV = days after vaccination.

**Figure 4 viruses-14-01916-f004:**
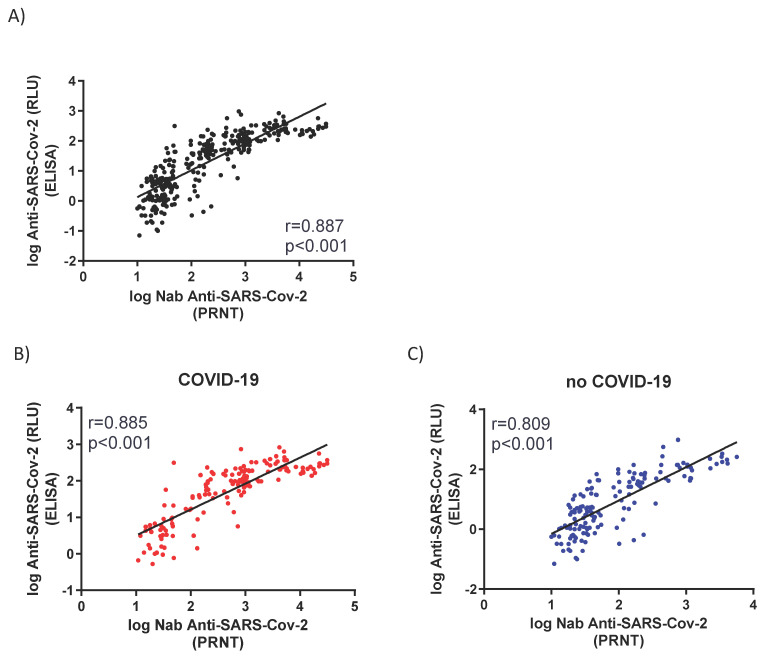
Correlations between neutralizing antibody (Nab) and total IgG. Graphics represent the correlation between anti-SARS-CoV-2 Nab and total IgG anti-RBD represented in RLUs. Panel (**A**) represents the correlation of all samples collected at 0, 7, 15, 90, and 120 DAV. Panels (**B**,**C**) represent correlations of values obtained at 120 DAV considering individuals without (noCOVID-19) or with (COVID-19) pre-vaccination SARS-CoV-2 infection, respectively. R and p values were obtained by Spearman correlations. DAV = days after vaccination. RLU = relative luminescence units.

**Figure 5 viruses-14-01916-f005:**
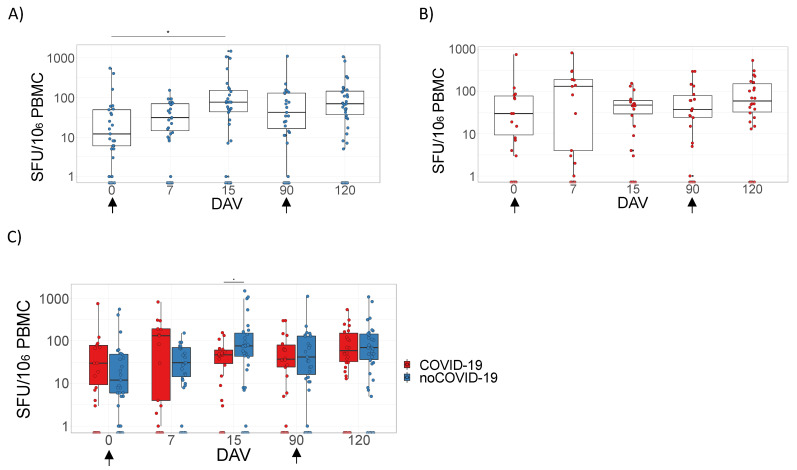
IFN-γ secretion after ChAdOx1 nCoV-19 vaccination. Panels (**A**–**C**) refer to samples from volunteers immunized with the prime–boost protocol for ChAdOx1 nCoV-19 on days 0 and 90 (indicated by arrows). Data are represented by box plots with the mean and interquartile intervals of IFN-γ spot forming units (SFU). Panels (**A**,**B**) represent comparisons between time points considering participants without pre-vaccination COVID-19 disease (noCOVID-19) (**A**) or with pre-vaccination COVID-19 (COVID-19) (**B**) using Kruskal–Wallis tests with Dunn’s multiple comparison post-hoc tests. Panel (**C**) represents comparisons between the noCOVID-19 and COVID-19 groups according to the time point analyzed using the Mann–Whitney test. The following levels denoted significance: * *p* < 0.05. Sample numbers vary according to follow-up segment loss and ELISpot technical exclusion criteria. DAV = days after vaccination.

**Figure 6 viruses-14-01916-f006:**
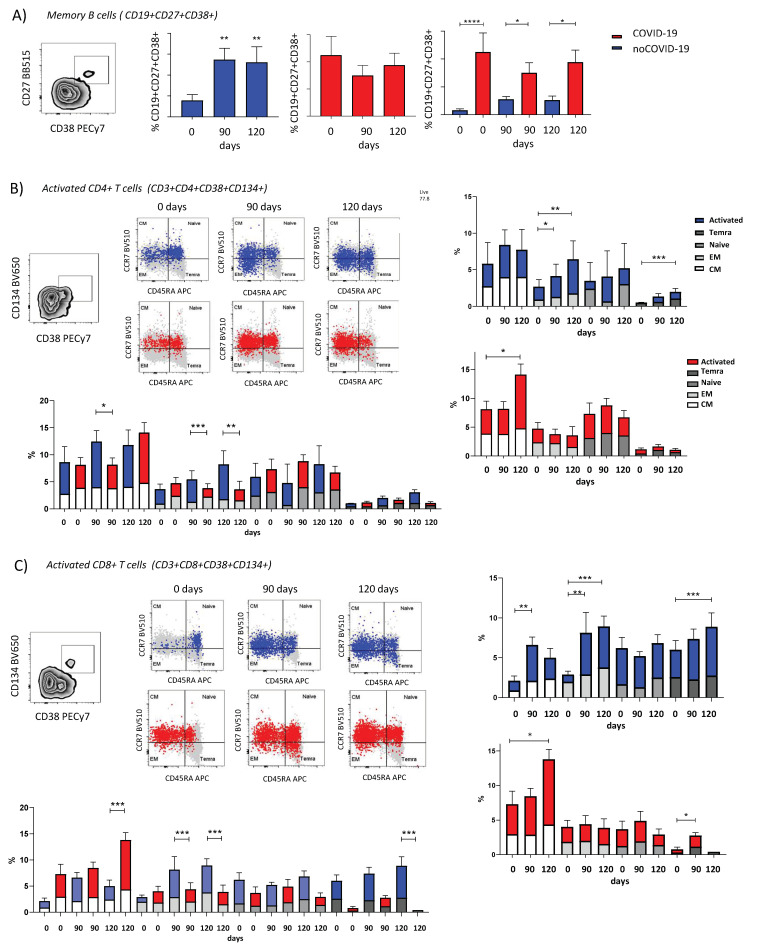
Immunophenotyping of memory B and T lymphocytes after ChAdOx1 nCoV-19 vaccination. The activated cells for the COVID-19 group are in red, and the activated cells for the no-COVID group are in blue for the B and T cell phenotypes. The profiles of (**A**) memory B cells (CD19+ CD27+ CD38+), (**B**) activated memory CD4+ T cells (CD3+ CD4+ CD38+ CD134+), (and (**C**) CD8+ T cells (CD3+ CD8+ CD38+ CD134+) and subpopulations of terminally differentiated effectors (Temra) (dark grey), naïve (grey), effector memory (EM) (light gray), and central memory (CM) (white) cells determined by CCR7 and CD45APC markers. Bar charts representing the mean and interquartile intervals of cellular populations comparing time points within groups and contrasting groups according to respective time points. Comparisons between time points were performed using Kruskal–Wallis tests with Dunn’s multiple comparison post-hoc tests. Comparisons were made between the noCOVID-19 and COVID-19 groups according to the time point analyzed using the Mann–Whitney tests. The following denoted significance levels: * *p* < 0.05, ** *p* < 0.01, *** *p* < 0.001, and **** *p* < 0.0005.

**Table 1 viruses-14-01916-t001:** Clinical and demographic features of enrolled participants.

	No COVID-19 (n = 58)	COVID-19 (n = 37)
**Age** (average ± SD)	39.1 ± 8.7	37.1 ± 9.6
**BMI** (average ± SD)	30 ± 20.5	28.1 ± 5.1
**Sex** (F/M)	39/19	24/13
**COVID-19 clinical classification** (asymptomatic/mild/severe)		7/27/3

Age given in years; BMI = body weight index [weight (kg)/ height^2^ (m)]; F = female; M = male.

**Table 2 viruses-14-01916-t002:** Techniques applied according to days after vaccination.

Parameter Analyzed	0 DAV (1st Dose)	7 DAV	15 DAV	30 DAV	90 DAV (2nd Dose)	120 DAV
Plasma D-Dimer	X	X	X			
Plasma total anti-spike IgG	X	X	X		X	X
Serum neutralizing antibodies	X	X	X	X	X	X
IFN-γ secretion by PBMC	X	X	X		X	X

DAV = days after vaccination.

## Data Availability

Data are available on request due to restrictions, e.g., privacy or ethics. The data presented in this study are available upon request from the corresponding author. The data are not publicly available due to ethical reasons.

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
