# Peer review of "Protective Immunity of COVID-19 Vaccination with ChAdOx1 nCoV-19 Following Previous SARS-CoV-2 Infection: A Humoral and Cellular Investigation"

_viruses, 2022, doi:10.3390/v14091916_

Round 1

Reviewer 1 Report

The authors investigated the humoral and cellular immune response from ChAadOx1 nCoV-19 vaccination in naive (noCOVID-19) and previously infected individuals (COVID-19). The manuscript is well-written and well-thought. 

The article can be improved with English language grammar mechanics and spelling.

Author Response

We would like to thank reviewer#1 for the gentle feedback. The English, including grammar mechanics and spelling, was deeply revised. In this way, we expect that edited manuscript achieves standards of the Viruses journal. Please find the certificate of proofreading in attach.

Reviewer 2 Report

Major comments:

1.) Wuhan SARS-CoV-2 spike protein gene, which Wuhan strain are authors referring to? Any accession number? 

2.) What is the rationale of using 60PFU for PRNT assay? 

3.) There are quite a number of VOI and VOC circulating globally while the type strain has already faded out. What is the significance of using the type strain instead of the current one for this study? It shouldn't be hard to find any new VOC from a country.

4.) What is the explanation with the IFN-γ secretion at 7DAV with high dispersion? Such dispersion is very significant when comparing with the noCOVID-19 group and what's the implication? 

5.) Use different coloring for the different immune cells, Terra/naive/EM and CM in figure 6. It is confusing. 

6.) Replace the figures with higher pixels. Wordings are blurry. 

7.) Where are the figure definitions for different coloring in figure 6D?

8.) How can the authors confirm if the number of time of COVID-19 infections experienced by the COVID19 group? Any chance if some of them have experienced a couple time of reinfections? Will that affect the result? 

Minor comments: 

1.) Suggest the authors to check the grammatical mistakes throughout the manuscript with native English speakers extensively. 

2.) Be specific on the types of vaccine. Live attenuated what? what DNA are authors referring to?

3.) "IFN-gamma quantification". 

4.) Rephrase "Many clinical trials using homologous prime-boost doses 8 to 12 weeks apart, analyzed samples from SARS-CoV-2 naïve individuals to investigate the cellular and humoral responses after vaccination by quantification of IFN-γ, total anti-spike and neutralizing antibodies, which are considered the standard analytes for SARS-CoV-2 vaccine efficacy" It is not organized. 

5.) These include.....humoral response,...

Author Response

Dear Dr. Elizabeth Yi,

Thank you for providing us an opportunity to revise our manuscript to the Viruses journal. We would like to thank the reviewers for their constructive criticism and valuable suggestions that have helped improve the manuscript. We have answered each query step by step, and the modifications have been highlighted (red) in the edited manuscript. Additionally, the editions have been referred to with a page number and respective line number in the revised manuscript. With the substantial edition, including a massive English revision, we believe to have adequately addressed the reviewer’s concerns. We hope that the revised manuscript is now suitable for publication in the Viruses.

Note of correction in the name of one author. The name of the author Ingrid Horbach Siciliano was wrong and is corrected in the edited version of the present manuscript.

Reviewer comments:

Reviewer #1

General comments

The authors investigated the humoral and cellular immune response from ChAadOx1 nCoV-19 vaccination in naive (noCOVID-19) and previously infected individuals (COVID-19). The manuscript is well-written and well-thought. 

The article can be improved with English language grammar mechanics and spelling.

R: We would like to thank reviewer#1 for the gentle feedback. The English, including grammar mechanics and spelling, was deeply revised. In this way, we expect that edited manuscript achieves standards of the Viruses journal. Please find the certificate of proofreading in attach.

Reviewer #2

Major comments:

1.) Wuhan SARS-CoV-2 spike protein gene, which Wuhan strain are authors referring to? Any accession number? 

R: Thank you for the important comment. The authors refer to the Wuhan SARS-CoV-2 spike protein gene that was codon-optimized for expression in human cell lines, GenBank accession number YP_009724390.1 (van Doremalen N et al. ChAdOx1 nCoV-19 vaccine prevents SARS-CoV-2 pneumonia in rhesus macaques. Nature. 2020 Oct;586(7830):578-582. doi: 10.1038/s41586-020-2608-y. Epub 2020 Jul 30. Erratum in: Nature. 2021 Feb;590(7844): E24. PMID: 32731258; PMCID: PMC8436420). GenBank accession number was included in the Introduction section (page 2, line 72) of the edited manuscript. In this sense, the information about SARS-CoV-2 clinical isolate used here in PRNT assay was also provided in the Methods section (page 4, line 165) of the edited manuscript.

“One of the vaccines created was ChAadOx1 nCoV-19 (AZD1222), developed at the University of Oxford by combining a codon-optimized full-length Wuhan SARS-CoV-2 spike protein gene (GenBank accession number YP_009724390.1) [1,2]”

“The SARS-CoV-2 virus was obtained from the inoculum of African green monkey Vero E6 cells with a clinical sample of a nasopharyngeal swab from a positive person for COVID-19, Rio de Janeiro. This isolate was kindly provided by the Laboratory of Respiratory Viruses and Measles at IOC/Fiocruz (SISGEN A994A37).”

2.) What is the rationale of using 60PFU for PRNT assay? 

R: Thank you for your question. The use of 60PFU/well was determined after PRNT standardization using different PFU amounts (50 to 120PFU). And this amount of PFU/well gives us homogeneous and individualized plaques, facilitating plaque counting and increasing the reliability of our data.

To further clarify this point, a brief explanation about the development of PRNT test was added to the edited manuscript (page 4, line 165):

“For the development of the PRNT, different plaque-forming units (PFU) amounts were tested (50 to 120 PFU). The amount of 60 PFU generated homogeneous and individualized plaques and was established as the standard for the assay (data not shown).”

3.) There are quite a number of VOI and VOC circulating globally while the type strain has already faded out. What is the significance of using the type strain instead of the current one for this study? It shouldn't be hard to find any new VOC from a country.

R: Thank you for your interesting comment.  Chibwana et al., 2022, published an elegant virological investigation demonstrating that AstraZeneca COVID-19 Vaccine elicits a protective broadly cross-reactive humoral response in a Malawian cohort previously infected with SARS-CoV-2 VOCs.  Unfortunately, development and validation of tests based of VOC were a challenge. Considering that previous works already characterized this AstraZeneca COVID-19 Vaccine cross-reactivity, in the present work we sought to characterize in general the influence of previous exposition to SARS-CoV-2 antigens in the human immunological humoral and cellular responses against the vaccine, independently of what SARS-CoV-2 VOC individuals were previously exposed.

In order to remark this limitation, we added a sentence in the last paragraph of discussion as following (page 13, line 449):

“Additionally, further investigations are necessary to characterize humoral and cellular responses against circulating VoCs, as well as the influence of previous exposure to multiple VoCs in immunological responses elicited by ChAadOx1 nCoV-19”

4.) (i) What is the explanation with the IFN-γ secretion at 7DAV with high dispersion? (ii)Such dispersion is very significant when comparing with the noCOVID-19 group and what's the implication? 

R: Thank you very much for your comment. (i) Our group observed the same high dispersion in early IFN-γ secretion (7DAV) after yellow fever vaccination, and we discovered an association of high/low levels with genetic background of the donors, specifically in two Single Nucleotide Polymorphisms (SNPs) in IFNG locus: rs2430561, rs2069718. Donors carrying alleles “A” in both SNPs presented higher early IFNG expression after yellow fever vaccination (Azamor et al., 2021). Hence, here we hypothesize that the cohort dispersion observed in IFN-γ secretion might be due to IFNG genetic background polymorphisms. (ii) The yellow fever vaccine is based on the live-attenuated replicative virus 17D/17DD that leads to a complex Th1/Th2 response, as observed in SARS-CoV-2 infection [3]. Herein, here our data suggest that the complex immunological responses elicited by previous SARS-CoV-2 exposition could lead to early and polarized IFN-γ response after vaccination.

To clarify these questions, we added the following sentences in bold type to the discussion topic (page 13, line 422):

However, some individuals with a previous COVID-19 history demonstrated a faster immune response at 7DAV, suggesting an influence of IFNG genetic background in vaccine immunogenicity. This genetically-regulated early IFN-γ response was already observed after immunization with the yellow fever (YF) vaccine 17DD, a replicative live-attenuated virus that elicits a complex Th1/Th2 response [4]. More efforts are necessary to further characterize the ChAadOx1 nCoV-19 response, although it is known that SARS-CoV-2 infections elicits a complex Th1/Th2 immunological response[3]. The need for balance in effector T cell response elicited by SARS-CoV-2 was already explored in multiple studies focusing on COVID-19 severity, demonstrating the role of lymphocyte reduction and exhaustion, and its relation to final disease outcome [5].”

5.) Use different coloring for the different immune cells, Terra/naive/EM and CM in figure 6. It is confusing. 

R: Thank you for your comment. We add different color to the bar graph and the information for each subset was included to the Figure 6 legend.

6.) Replace the figures with higher pixels. Wordings are blurry. 

Thank you for your comment. All figures were replaced by a higher resolution image. Specifically figures 4 and 6 were re-edited to improve quality. Please find the improved figures in the edited version of this manuscript, and in attach.

7.) Where are the figure definitions for different coloring in figure 6D?

R: Thank you for your comment. We include the information in the legend.

Figure 6. Immunophenotyping of memory B and T lymphocytes after ChAdOx1 nCoV-19 vaccination. Immunophenotyping gating strategy for B and T memory cells (A). The activated cells for COVID-19 group are in red, and the activated cells for no-COVID group are in blue color for B and T cells phenotypes. Profile of memory B cells (CD19+CD27+CD38+), (B), activated memory CD4+ cells (CD3+CD4+CD38+CD134+) (C) and CD8+ (CD3+CD8+CD38+CD134+) (D) T cells and subpopulations of terminally differentiated effectors (Temra) (dark grey), Naïve (grey), Effector Memory (EM) (light grey), and Central Memory (CM) cells (white) determined by CCR7 and CD45APC markers. Bar charts representing the mean and interquartile intervals of cellular populations comparing time points withing groups and contrasting groups according to respective time points. Comparisons between timepoints using Kruskal-Wallis with Dunn's multiple comparison post-test.  Comparisons between noCOVID and COVID-19 groups according to the time point analyzed using the Man-Whitney test. Considering *p<0.05, **p<0.01, and ***p<0.001.

8.) How can the authors confirm if the number of time of COVID-19 infections experienced by the COVID19 group? Any chance if some of them have experienced a couple time of reinfections? Will that affect the result? 

R: Thank you for your comment. From the experience of your group, it has already been observed that prior exposure to SARS-CoV-2 can influence the post-vaccination humoral immune response by increasing the titer of neutralizing antibodies compared to naive individuals. However, unfortunately here we could not determine how many times a participant has been infected with the virus, or how the number of previous infections may influence the immune response.

Due to the increasing number of cases of infection with multiple VoCs, we consider it a limitation in our study. To further state this limitation, we added a sentence in the last paragraph of discussion as following (page 13, line 449):

“Additionally, further investigations are necessary to characterize humoral and cellular responses against circulating VoCs elicited by the ChAadOx1 nCoV-19 vaccine, as well as the influence of previous exposure to multiple VoCs in immunological responses elicited by ChAadOx1 nCoV-19”.

Minor comments: 

1.) Suggest the authors to check the grammatical mistakes throughout the manuscript with native English speakers extensively. 

R: Thank you for your comment. The English, including grammar mechanics and spelling, was deeply revised by a professional service (please find in attach the certificate of proofreading). In this way, we expect that edited manuscript achieves standards of the Viruses journal.

2.) Be specific on the types of vaccine. Live attenuated what? what DNA are authors referring to?

R: Thank you for your comment. The vaccines mentioned were better specified in the text as following (page 2, line 72):

“The knowledge acquired with previous emergence of SARS-CoV-1 and MERS-CoV, helped in the rapid development of SARS-CoV-2 vaccines. A diversity of vaccines were developed including a DNA-based by Yu et al., based on live attenuated YF17D-vector by Sanchez-Felipe, as well as the mRNA-based by Pfized-BioNTech (BNT162b2) and Moderna (mRNA-1273) [6–9]”

3.) "IFN-gamma quantification". 

R: Please find the response bellow in item 5.

4.) Rephrase "Many clinical trials using homologous prime-boost doses 8 to 12 weeks apart, analyzed samples from SARS-CoV-2 naïve individuals to investigate the cellular and humoral responses after vaccination by quantification of IFN-γ, total anti-spike and neutralizing antibodies, which are considered the standard analytes for SARS-CoV-2 vaccine efficacy" It is not organized. 

R: Please find the response bellow in item 5.

5.) These include.....humoral response,...

R: Thank you for your contribution, the sentence was rephased. Please find the rephased text bellow that responds to items 3,4 and 5 (page2, line 76):

“Many clinical trials using homologous prime-boost doses 8 to 12 weeks apart, analyzed samples from SARS-CoV-2 naïve individuals to investigate their cellular and humoral responses after vaccination. These investigations included quantitation of IFN-γ secreted cells, total anti-spike and neutralizing antibodies, which are considered the standard analytes for SARS-CoV-2 vaccine efficacy [10].”

References:

  1. van Doremalen, N.; Lambe, T.; Spencer, A.; Belij-Rammerstorfer, S.; Purushotham, J.N.; Port, J.R.; Avanzato, V.A.; Bushmaker, T.; Flaxman, A.; Ulaszewska, M.; et al. ChAdOx1 NCoV-19 Vaccine Prevents SARS-CoV-2 Pneumonia in Rhesus Macaques. Nature 2020, 586, 578–582, doi:10.1038/s41586-020-2608-y.
  2. van Doremalen, N.; Lambe, T.; Spencer, A.; Belij-Rammerstorfer, S.; Purushotham, J.N.; Port, J.R.; Avanzato, V.A.; Bushmaker, T.; Flaxman, A.; Ulaszewska, M.; et al. ChAdOx1 NCoV-19 Vaccine Prevents SARS-CoV-2 Pneumonia in Rhesus Macaques. Nature 2020, 586, 578–582, doi:10.1038/s41586-020-2608-y.
  3. Ravindran, R.; McReynolds, C.; Yang, J.; Hammock, B.D.; Ikram, A.; Ali, A.; Bashir, A.; Zohra, T.; Chang, W.L.W.; Hartigan-O’Connor, D.J.; et al. Immune Response Dynamics in COVID-19 Patients to SARS-CoV-2 and Other Human Coronaviruses. PLoS ONE 2021, 16, e0254367, doi:10.1371/journal.pone.0254367.
  4. Azamor, T.; da Silva, A.M.V.; Melgaço, J.G.; dos Santos, A.P.; Xavier-Carvalho, C.; Alvarado-Arnez, L.E.; Batista-Silva, L.R.; de Souza Matos, D.C.; Bayma, C.; Missailidis, S.; et al. Activation of an Effective Immune Response after Yellow Fever Vaccination Is Associated with the Genetic Background and Early Response of IFN-γ and CLEC5A. Viruses 2021, 13, 96, doi:10.3390/v13010096.
  5. Azkur, A.K.; Akdis, M.; Azkur, D.; Sokolowska, M.; van de Veen, W.; Brüggen, M.-C.; O’Mahony, L.; Gao, Y.; Nadeau, K.; Akdis, C.A. Immune Response to SARS-CoV-2 and Mechanisms of Immunopathological Changes in COVID-19. Allergy 2020, 75, 1564–1581, doi:10.1111/all.14364.
  6. Sanchez-Felipe, L.; Vercruysse, T.; Sharma, S.; Ma, J.; Lemmens, V.; Van Looveren, D.; Arkalagud Javarappa, M.P.; Boudewijns, R.; Malengier-Devlies, B.; Liesenborghs, L.; et al. A Single-Dose Live-Attenuated YF17D-Vectored SARS-CoV-2 Vaccine Candidate. Nature 2021, 590, 320–325, doi:10.1038/s41586-020-3035-9.
  7. Yu, J.; Tostanoski, L.H.; Peter, L.; Mercado, N.B.; McMahan, K.; Mahrokhian, S.H.; Nkolola, J.P.; Liu, J.; Li, Z.; Chandrashekar, A.; et al. DNA Vaccine Protection against SARS-CoV-2 in Rhesus Macaques. Science 2020, 369, 806–811, doi:10.1126/science.abc6284.
  8. Baden, L.R.; El Sahly, H.M.; Essink, B.; Kotloff, K.; Frey, S.; Novak, R.; Diemert, D.; Spector, S.A.; Rouphael, N.; Creech, C.B.; et al. Efficacy and Safety of the MRNA-1273 SARS-CoV-2 Vaccine. N Engl J Med 2021, 384, 403–416, doi:10.1056/NEJMoa2035389.
  9. Polack, F.P.; Thomas, S.J.; Kitchin, N.; Absalon, J.; Gurtman, A.; Lockhart, S.; Perez, J.L.; Pérez Marc, G.; Moreira, E.D.; Zerbini, C.; et al. Safety and Efficacy of the BNT162b2 MRNA Covid-19 Vaccine. N Engl J Med 2020, 383, 2603–2615, doi:10.1056/NEJMoa2034577.
  10. Grigoryan, L.; Pulendran, B. The Immunology of SARS-CoV-2 Infections and Vaccines. Seminars in Immunology 2020, 50, 101422, doi:10.1016/j.smim.2020.101422.

Best Regards,

Tamiris Azamor

Round 2

Reviewer 2 Report

Thanks for the replies. Authors have tried their best to answer the questions appropriately.